# A conserved membrane protein negatively regulates Mce1 complexes in mycobacteria

Yushu Chen [1], Yuchun Wang [1] & Shu-Sin Chng [1,2] ✉

Tuberculosis continues to pose a serious threat to global health. *Mycobacterium tuberculosis*, the causative agent of tuberculosis, is an intracellular pathogen that relies on various mechanisms to survive and persist within the host. Among their many virulence factors, mycobacteria encode Mce systems. Some of these systems are implicated in lipid uptake, but the molecular basis for Mce function(s) is poorly understood. To gain insights into the composition and architecture of Mce systems, we characterized the putative Mce1 complex involved in fatty acid transport. We show that the Mce1 system in *Mycobacterium smegmatis* comprises a canonical ATP-binding cassette transporter associated with distinct heterohexameric assemblies of substrate-binding proteins. Furthermore, we establish that the conserved membrane protein Mce1N negatively regulates Mce1 function via a unique mechanism involving blocking transporter assembly. Our work offers a molecular understanding of Mce complexes, sheds light on mycobacterial lipid metabolism and its regulation, and informs future anti-mycobacterial strategies.

Tuberculosis (TB) has been one of the top killers among single infectious diseases in humans, alone claiming 1.4 million lives out of 10.6 million new cases in 2021[1]. Apart from the active cases, it is estimated that one-quarter of the world population has latent TB, which may progress into active infections[2]. TB is primarily caused by *Mycobacterium tuberculosis*, an organism known for its ability to adapt to the host environment and modulate the host immune system[3]. Its success as a pathogen can be owed to diverse virulence factors encoded in the genome of *M. tuberculosis*. In particular, many of these virulence factors promote host adaptation by contributing to the acquisition and catabolism of important nutrients, including metals, sugars and lipids[4].

*mce* operons were initially recognized as virulence factors in *M. tuberculosis*[5] and later found to be widespread throughout the *Mycobacterium* genus[6]. *M. tuberculosis* contains four *mce* operons, individually important determinants of virulence and collectively critical for pathogenesis[7–10]. *Mycobacterium smegmatis*, an environmental microorganism, contains six *mce* operons, while *Mycobacterium leprae*, the causative agent of leprosy, has only one such operon (*mce1*) in its highly degenerate genome[6]. In general, an *mce* operon consists of eight genes, two *yrbE* (e.g., *yrbE1A/B*) and six *mce* genes (e.g., *mce1A–F*),

with different numbers of accessory genes[6] that may be required for function[11,12] (e.g., *mam1A–D* after the *mce1* operon). Furthermore, a gene encoding a transcriptional regulator (e.g., *mce1R*) can be found upstream of some *mce* operons[6,13,14] (Fig. 1a and Supplementary Fig. 1). The *mce1* and *mce4* operons have been implicated in fatty acid and cholesterol uptake, respectively[9,11,15]. Our understanding of how these operons facilitate lipid uptake is limited.

It is believed that *mce* operons encode ATP-binding cassette (ABC) transporters that are analogous to the MlaFEDB complex in Gram-negative bacteria[6,16], which is important for retrograde transport of phospholipids[17,18]. In MlaFEDB, MlaF, MlaE and MlaD represent the nucleotide-binding domains (NBDs), transmembrane domains (TMDs), and substrate-binding proteins (SBPs), respectively[19,20]. In Mce systems, YrbEA/Bs and MceA–Fs would form the corresponding TMDs and SBPs based on homology; here, MceG, a protein not encoded within *mce* operons, is thought to constitute the NBDs that energize all these putative transporters[21,22] (Fig. 1b). MlaD and MceA–F all contain a single mammalian cell entry (MCE) domain. Interestingly, MlaD and other MCE-domain proteins, such as PqiB and YebT/LetB, have been shown to form homohexamers that provide a hydrophobic tunnel each, presumably for lipid substrate transport[19,20,23,24]. The fact that

[1]Department of Chemistry, National University of Singapore, Singapore 117543, Singapore. [2]Singapore Center for Environmental Life Sciences Engineering, National University of Singapore (SCELSE-NUS), Singapore 117456, Singapore. ✉e-mail: chmchngs@nus.edu.sg

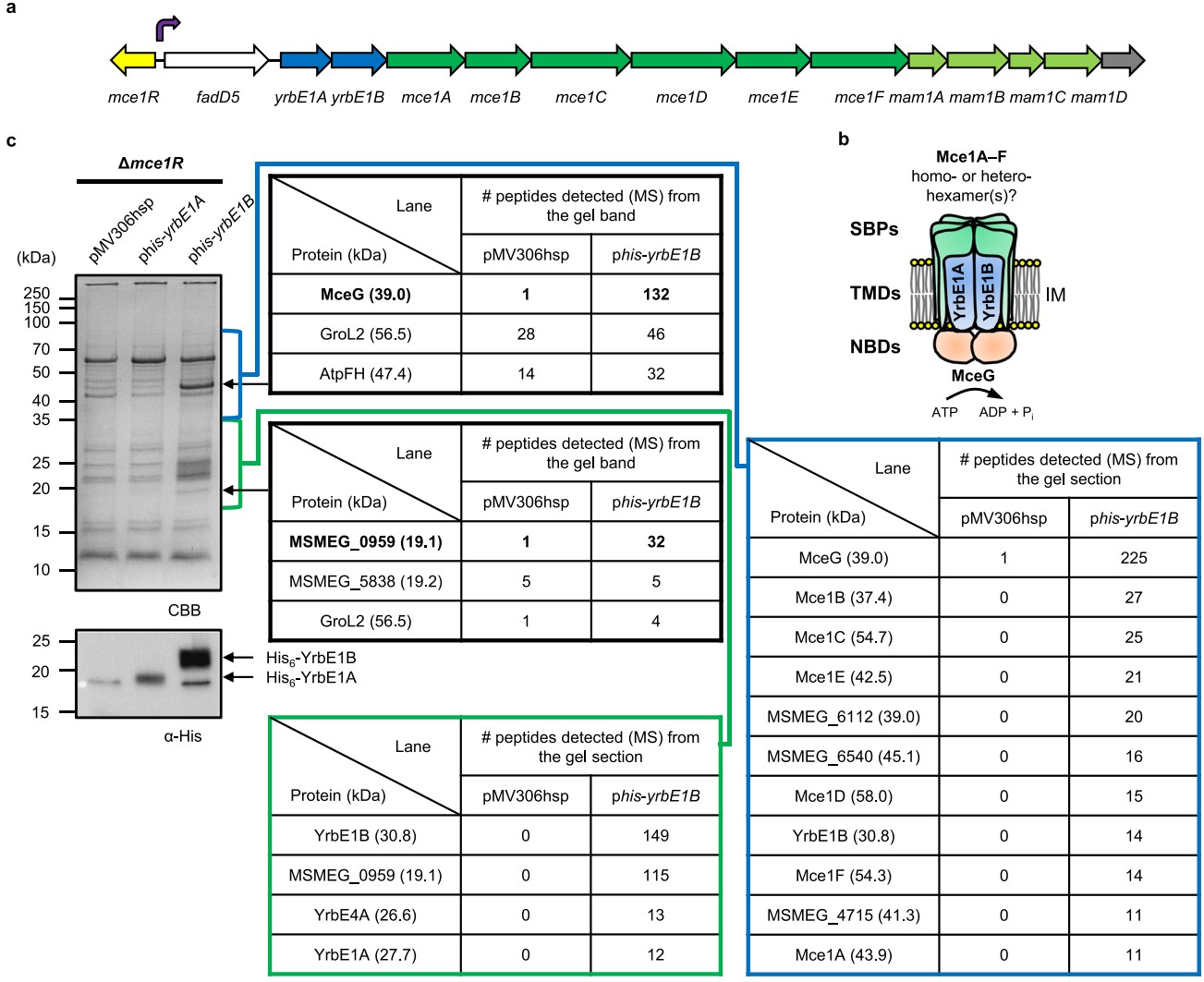

**Fig. 1 | Affinity purification with His₆-YrbE1B defines the composition of Mce1 complexes. a** A schematic representation of the *mce1* genetic loci in *M. smegmatis*. Each straight arrow represents an individual gene. The lengths of the arrows are scaled to the lengths of the genes. The gene encoding a putative transcriptional regulator of the *mce1* operon is colored in yellow. A gene encoding a putative acyl-CoA synthase is colored in white. *yrbE* genes are colored in blue, *mce* genes dark green, *mam* genes light green and unrelated genes gray. The promoter position of the *mce1* operon is annotated with a turning arrow in purple. **b** A basic model of the putative Mce1 complex based on current knowledge. SBP (Mce1A–F) substrate-binding protein, TMD (YrbE1A/B) transmembrane domain, NBD (MceG) nucleotide-binding domain, IM inner membrane. **c** SDS-PAGE and α-His immunoblot analyses

of proteins affinity-purified from *M. smegmatis* Δ*mce1R* cells expressing His-tagged YrbE1A (p*his-yrbE1A*) or YrbE1B (p*his-yrbE1B*). pMV306hsp was used as the empty vector control. For protein identification from gel bands, the top three proteins detected in the p*his-yrbE1B* lane are listed. The protein assigned to the band and the corresponding numbers of peptides detected are highlighted in bold. For protein identification from gel sections (blue: 35–90 kDa, green: 18–35 kDa), proteins with more than ten total peptides detected and significantly enriched in the p*his-yrbE1B* lane are listed. The uncropped gel, blot and full protein identification lists are in Source Data. The gel and the blot are representative of three independent experiments. CBB Coomassie brilliant blue, MS mass spectrometry.

*mce* operons encode six Mce proteins MceA–F hints at the possible formation of heterohexameric MCE assemblies within each putative transporter. Despite similarities to MlaFEDB, however, there has been a lack of biochemical studies on mycobacterial Mce systems, and their exact compositions and overall architectures remain unknown.

Here, we biochemically characterized the putative Mce1 complex in *M. smegmatis*. We show that this system is important for fatty acid uptake, akin to the *M. tuberculosis* Mce1 complex[11,15]. We demonstrate that Mce1 is an ABC transporter comprising a core complex of YrbE1A/B-MceG, associated with a likely heterohexamer of Mce1A–F. We reveal that MSMEG_6540 is a functional homolog that can replace Mce1A in the complex and, in fact, contributes more significantly to the fatty acid uptake function of Mce1. Furthermore, we identify MSMEG_0959, herein renamed to Mce1N, as a negative regulator of the Mce1 system; it prevents MceG from associating with the complex via competitive

binding to YrbE1B, thereby modulating transport activity. Our work provides critical insights into the architecture, function and regulation of Mce systems in mycobacteria.

## Results

### Key proteins encoded in the *mce1* operon co-purify as a complex

To probe the composition of the putative Mce1 complex, we expressed His-tagged YrbE1A or YrbE1B in *M. smegmatis* using an integrative plasmid and conducted affinity purification following cell lysis and detergent extraction from total membranes. This was done in a Δ*mce1R* strain, where the *mce1* operon is presumably upregulated[13]. We found that His-tagged YrbE1B, but not YrbE1A, pulled down two distinct protein bands at ~45 and ~19 kDa (Fig. 1c). Tandem mass spectrometry (MS/MS) sequencing of the excised bands indicated that they were MceG (39.0 kDa) and a protein of unknown function,

MSMEG_0959 (19.1 kDa), respectively. We did not observe additional bands corresponding to other expected members of the Mce1 complex, possibly due to low cellular levels or instability of the complex ex vivo. To detect potential low-abundance proteins specifically co-purified with His-tagged YrbE1B, we directly subjected entire gel lanes divided into two sections to MS/MS protein identification. Remarkably, this facilitated the detection of all other putative Mce1 components, including YrbE1A and Mce1A–F, similar to a recent report identifying YrbE4A/B, Mce4A–F and MceG as components of the Mce4 complex[12]. Interestingly, a seventh MCE-domain protein MSMEG_6540[6] was also co-purified with His₆-YrbE1B, while none of the putative Mce-associated membrane proteins (Mam1A–D and OmamA/B[25]) or LucA[11] was identified. Our data suggest that the Mce1 complex likely comprises the core ABC transporter YrbE1A/B-MceG associated with a presumed heterohexamer of MCE-domain proteins.

## MSMEG_6540 and Mce1A are components of distinct Mce1 complexes

MSMEG_6540 shares 80% primary sequence identity with Mce1A, compared to less than 20% with other MCE-domain proteins in the Mce1 complex and ~25% with MceA proteins in other putative *M. smegmatis* Mce complexes (Supplementary Table 1). Another Mce1A paralog MSMEG_5818[6], which is only 60% identical, was not co-purified with His-tagged YrbE1B (Fig. 1c). This suggests that MSMEG_6540 could be a functional homolog of Mce1A. We also hypothesized that MSMEG_6540 and Mce1A form heterohexamers with Mce1B–F in distinct complexes. To test these ideas, we examined the effects of removing each MCE-domain protein on fatty acid uptake, a key function of the Mce1 complex[11,15]. Deletion of *yrbE1A/B* or *mceG*, either expected to abolish Mce1 function, resulted in significantly reduced [¹⁴C]-palmitic acid uptake in cells (Fig. 2a and Supplementary Fig. 2). We established that [¹⁴C]-palmitic acid uptake was also reduced to the baseline, i.e., at levels of Δ*yrbE1A/B* cells, in individual Δ*mce1B*, Δ*mce1C*, Δ*mce1D*, Δ*mce1E* or Δ*mce1F* mutants. The Δ*mce1C* and Δ*mce1D* strains could be complemented by ectopic expression of the deleted gene alone (i.e., no polar effects) (Fig. 2b). The Δ*mce1B*, Δ*mce1E* or Δ*mce1F* mutants could be fully complemented only when the deleted gene was expressed together with downstream genes, but not when the latter alone was expressed (Fig. 2c–e). Therefore, each of Mce1B–F should be present in all Mce1 complexes and required for fatty acid uptake function.

Unexpectedly, removing Mce1A did not abolish, yet somehow gave increased [¹⁴C]-palmitic acid uptake (Fig. 2a). In contrast, the uptake was significantly less efficient in cells lacking *MSMEG_6540* and only further reduced to the baseline in the Δ*mce1A*Δ*6540* double mutant (Fig. 2a). We showed that expressing *mce1A* in the double mutant increased palmitic acid uptake slightly, to levels similar to the Δ*MSMEG_6540* strain, while expressing *MSMEG_6540* brought uptake to levels higher than wild-type (WT) cells, akin to the Δ*mce1A* strain (Fig. 2f). Our results establish that in the context of Mce1 complexes in *M. smegmatis*, MSMEG_6540 plays the major role in mediating fatty acid uptake. Interestingly, expressing extra Mce1A in WT cells decreased palmitic acid uptake in a Mce1-dependent manner (Supplementary Fig. 3). Hence, the more Mce1A there is, the lower the level of Mce1 complexes containing MSMEG_6540, and vice versa; this is consistent with the higher palmitic acid uptake observed in the Δ*mce1A* strain (Fig. 2a). Taken together, our data indicate that Mce1A and MSMEG_6540 compete to form distinct heterohexamers together with Mce1B–F, with the latter Mce1 complex playing a more important role in fatty acid uptake in *M. smegmatis*. While they can contribute slightly to fatty acid uptake, it is likely that Mce1A-containing complexes have a distinct role, perhaps in the uptake of other lipid substrates.

## YrbE1A/B-MceG constitute the core ABC transporter

We attempted but failed to over-express the *mce1* operon along with *mceG* in *E. coli*. Expression levels of Mce1A–F were very low, which

precluded purification of any stable Mce1 complex. Instead, we were able to isolate the core ABC transporter when we co-expressed YrbE1A, His-tagged YrbE1B and MceG. This complex eluted as a single peak on size exclusion chromatography (Fig. 3a); we showed that the purified complex exhibits ATP hydrolytic activity (Fig. 3b), implying proper assembly of YrbE1A/B-MceG. Interestingly, we found that His-tagged YrbE1B alone could pull down MceG when co-expressed in *E. coli*. In contrast, His-tagged YrbE1A could only co-purify with MceG in the presence of YrbE1B, indicating that the TMD-NBD interactions are mainly mediated through YrbE1B (Fig. 3c). Consistent with this idea, an AlphaFold2 structural model of the YrbE1A/B-MceG complex[26–28], which resembles the homologous MlaFE structure from Gram-negative bacteria (RMSD = 2.934 Å)[29], presented a smaller buried area between MceG and YrbE1A (699 Å²) compared to YrbE1B (837 Å²) (Supplementary Fig. 4 and Supplementary Table 2). We conclude that the TMD and NBD proteins of the Mce1 system interact asymmetrically to form a stable functional ABC transporter.

## MSMEG_0959 is a negative regulator of Mce1

In addition to MceG, MSMEG_0959 was another protein abundantly co-purified with His-tagged YrbE1B expressed in *M. smegmatis*. MSMEG_0959 is a putative conserved membrane protein with unknown function; it contains two predicted transmembrane helices (TM1: W39–G59; TM2: L65–T90) and a C-terminal cytoplasmic domain (S91–P178) (Supplementary Fig. 5a). We initially hypothesized that it might function to stabilize the Mce1 complex, analogous to the role of the cytoplasmic protein MlaB in the Gram-negative MlaFEDB complex[19]. To test this idea, we attempted to isolate a stable complex containing YrbE1A/B, MceG and MSMEG_0959. To our surprise, when these four proteins were co-expressed and purified from *E. coli*, only MSMEG_0959 was co-purified with His-tagged YrbE1B (Fig. 4a). We detected much less YrbE1A and essentially no MceG co-purified, suggesting that MSMEG_0959 may destabilize the Mce1 complex instead. In support of this notion, when MSMEG_0959 was over-expressed in *M. smegmatis*, much less MceG was co-purified with His-tagged YrbE1B (Fig. 4c). We conclude that MSMEG_0959 may disrupt YrbE1A/B-MceG interactions.

Since TMD-NBD interactions within the Mce1 complex are largely mediated through YrbE1B, we presumed that MSMEG_0959 directly interacts with YrbE1B. Indeed, His-tagged YrbE1B alone, but not YrbE1A, pulled down MSMEG_0959 when co-expressed in *E. coli*, consistent with affinity purification results in *M. smegmatis* (Fig. 1c, Supplementary Fig. 6a, b). We generated an AlphaFold2 structural model comprising YrbE1B and MSMEG_0959[26–28], predicting possible interactions between the transmembrane domain of MSMEG_0959 and that of YrbE1B opposite its dimerization surface with YrbE1A (Fig. 4d). To test the validity of this model, we introduced cysteine pairs across potential contact sites in the transmembrane regions of the two proteins, and looked for disulfide bond formation when these variants are expressed in *E. coli*. Five pairs of cysteines at the periplasmic side of the putative interface enabled crosslinking between YrbE1B and MSMEG_0959 (Fig. 5), lending confidence to the predicted interaction mode. The AlphaFold2 model revealed that an unstructured loop in the cytoplasmic domain of MSMEG_0959 may contact a short helix of YrbE1B sitting at the membrane-water boundary. This helix may be a determinant of the interaction between YrbE1B and MSMEG_0959 since the amino acid sequence of this region is very different from that in YrbE1A (Supplementary Fig. 6c, d). Mutating part of this helix (RLVAEIGMGT) to a flexible loop (GGSSG) did not affect the stability of the YrbE1A/B-MceG complex; however, this mutation prevented co-purification of MSMEG_0959 with His-tagged YrbE1B from *E. coli*, at the same time alleviating the destabilizing effect of MSMEG_0959 on YrbE1A/B-MceG interactions (Fig. 4a). Notably, overlay of the models of YrbE1B-MSMEG_0959 and YrbE1A/B-MceG highlighted substantial steric clashes between MceG and the cytoplasmic domain of MSMEG_0959

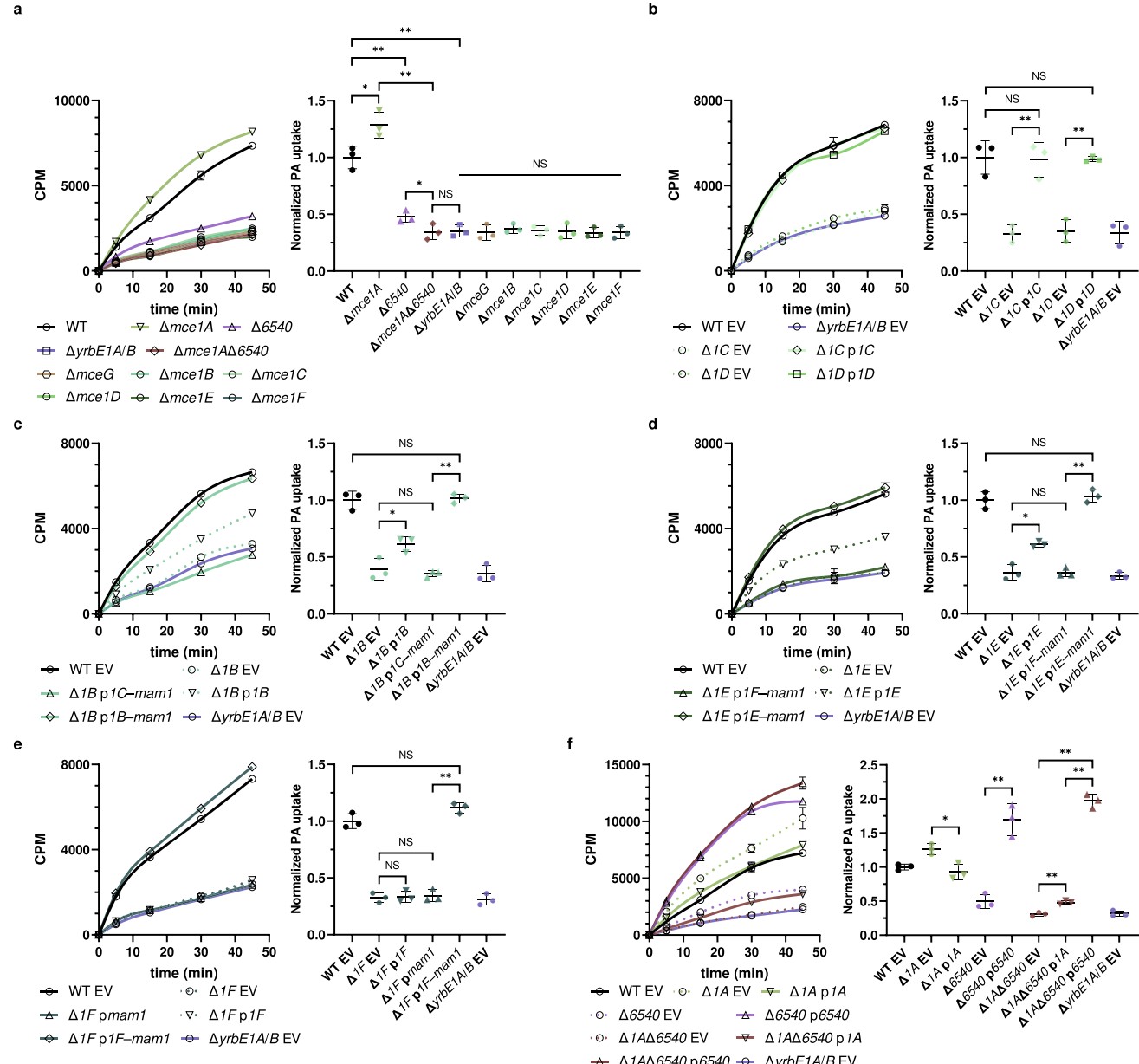

**Fig. 2 | MSMEG_6540 and Mce1A are redundant components that likely form heterohexamers with Mce1B–F in Mce1 complexes. a–f** [$^{14}$C]-palmitic acid uptake profiles and rates of indicated *M. smegmatis* strains. Each uptake profile (left) shows accumulated radioactivity counts in cells over time and is representative of at least three independent experiments. Each data point (mean ± standard deviation) represents results from three technical replicates. CPM count per minute. Uptake rates (right) are quantified based on [$^{14}$C]-palmitic acid levels after 30-min incubation. The uptake of individual strains is normalized to that of wild-type (WT) cells or WT cells harboring the empty vector (WT EV). Mean ± standard deviation of three biological replicates is shown for each group. In (**b–f**), different strains in (**a**) are complemented via the expression of indicated genes from a plasmid (annotated as p followed by the abbreviated gene name). *MSMEG_6540* and *mce1A–F* genes are annotated as *6540* and *1A–F*, respectively. EV empty vector (pJEB402). One-way repeated measures ANOVA: NS, not significant; * $p < 0.05$; ** $p < 0.01$. In (**a**): WT vs Δ*mce1A* ($p = 0.014$), WT vs Δ*6540* ($p = 0.0051$), WT vs Δ*yrbE1A/B* ($p = 0.0062$),

Δ*mce1A*Δ*6540* vs Δ*yrbE1A/B* ($p = 0.59$), Δ*mce1A* vs Δ*mce1A*Δ*6540* ($p = 0.0022$), Δ*6540* vs Δ*mce1A*Δ*6540* ($p = 0.029$). There is no significant difference between the Δ*yrbE1A/B* strain and any other strain under the NS line (from left to right, $p = 0.54$, 0.097, 0.91, 0.88, 0.16 and 0.57, respectively). In (**b**): Δ*1C* EV vs Δ*1C* p*1C* ($p = 0.0051$), WT EV vs Δ*1C* p*1C* ($p = 0.28$), Δ*1D* EV vs Δ*1D* p*1D* ($p = 0.0049$), WT EV vs Δ*1D* p*1D* ($p = 0.86$). In (**c**): Δ*1B* EV vs Δ*1B* p*1B* ($p = 0.032$), Δ*1B* EV vs Δ*1B* p*1C-mam1* ($p = 0.51$), Δ*1B* p*1C-mam1* vs Δ*1B* p*1B-mam1* ($p = 0.0019$), WT EV vs Δ*1B* p*1B-mam1* ($p = 0.81$). In (**d**): Δ*1E* EV vs Δ*1E* p*1E* ($p = 0.014$), Δ*1E* EV vs Δ*1E* p*1F-mam1* ($p = 0.97$), Δ*1E* p*1F-mam1* vs Δ*1E* p*1E-mam1* ($p = 0.0037$), WT EV vs Δ*1E* p*1E-mam1* ($p = 0.45$). In (**e**): Δ*1F* EV vs Δ*1F* p*1F* ($p = 0.50$), Δ*1F* EV vs Δ*1F* p*mam1* ($p = 0.44$), Δ*1F* p*mam1* vs Δ*1F* p*1F-mam1* ($p = 0.0017$), WT EV vs Δ*1F* p*1F-mam1* ($p = 0.12$). In (**f**): Δ*1A* EV vs Δ*1A* p*1A* ($p = 0.020$), Δ*6540* EV vs Δ*6540* p*6540* ($p = 0.0054$), Δ*1A*Δ*6540* EV vs Δ*1A*Δ*6540* p*1A* ($p = 0.0073$), Δ*1A*Δ*6540* EV vs Δ*1A*Δ*6540* p*6540* ($p = 0.0019$), Δ*1A*Δ*6540* p*1A* vs Δ*1A*Δ*6540* p*6540* ($p = 0.0019$). The raw data underlying the figures are provided in Source Data.

(Fig. 4d), indicating that their respective interactions with YrbE1B are likely competitive and mutually exclusive. We also generated an MSMEG_0959 variant (MSMEG_0959$_{AGG3P}$) where the cytoplasmic loop that appears to interact with YrbE1B is modified ($^{131}$ALGELTG to PLPELTP) to impact overall conformation. Intriguingly, this variant still

interacts with YrbE1B but is no longer able to compete away MceG when co-expressed in either *E. coli* (Fig. 4b) or M. *smegmatis* (Fig. 4c). Taken together, our results establish that MSMEG_0959 specifically binds YrbE1B and prevents association of the NBD (MceG) to this ABC transporter.

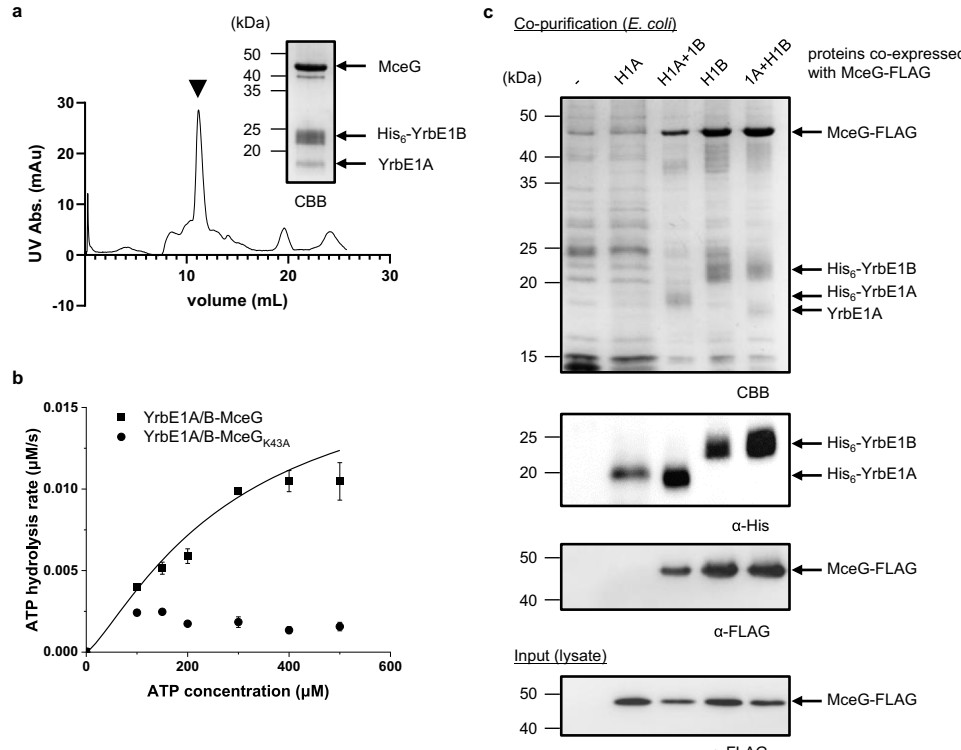

**Fig. 3 | YrbE1A/B-MceG form a stable asymmetric complex with ATPase activity.**
**a** Size exclusion chromatographic profile of the purified complex comprising YrbE1A, His$_6$-YrbE1B and MceG. SDS-PAGE analysis of the peak corresponding to the complex (black arrowhead) is shown. Abs. absorbance. **b** Enzyme-coupled ATPase assay of 0.1 µM YrbE1A/B-MceG and YrbE1A/B-MceG$_{K43A}$ in detergent micelles at 37 °C. Average ATP hydrolysis rates from three technical triplicates were plotted against ATP concentrations and fitted to an expanded Michaelis-Menten equation that includes a term for Hill coefficient (n). YrbE1A/B-MceG: K$_m$ = 284 ± 311 µM, $k_{cat}$ = 0.184 ± 0.125 µM ATP s$^{-1}$/µM complex, $n$ = 1.28 ± 0.57. The data for the mutant

failed to fit the equation with significant confidence. Each data point is shown as mean ± standard deviation. **c** SDS-PAGE and α-His/FLAG immunoblot analyses of proteins affinity-purified from *E. coli* cells over-expressing FLAG-tagged MceG together with either His$_6$-YrbE1A (H1A) with/without YrbE1B (1B) or His$_6$-YrbE1B (H1B) with/without YrbE1A (1A). CBB, Coomassie brilliant blue. For (**a**–**c**), the experiments were repeated three times independently with similar results. The uncropped gels, blots, and raw data underlying (**a**) and (**b**) are provided in Source Data.

Our biochemical experiments suggest that MSMEG_0959 may be a negative regulator of the Mce1 complex. Consistent with this idea, MSMEG_0959 over-expression inhibited [$^{14}$C]-palmitic acid uptake in WT cells, but not the residual uptake observed in Δ*yrbE1A/B* cells (Fig. 4e). In contrast, over-expression of MSMEG_0959$_{AGG3P}$ did not inhibit [$^{14}$C]-palmitic acid uptake at all (Fig. 4e), in line with its inability to displace MceG (Fig. 4c). Furthermore, we found that over-expression of Rv0513, the *M. tuberculosis* homolog of MSMEG_0959, reduced the amount of MceG associated with YrbE1B (Fig. 4c) in *M. smegmatis*, and slowed down [$^{14}$C]-palmitic acid uptake (Fig. 4e). We therefore conclude that MSMEG_0959/Rv0513, herein renamed to Mce1N (Mce1 <u>n</u>egative regulator), inhibits the fatty acid uptake function of the Mce1 transporter(s) via a conserved mechanism that involves prevention of YrbE1B-MceG interaction.

## Discussion

In this study, we set out to characterize the Mce1 complex in *M. smegmatis*. We have shown that the Mce1 complexes likely comprise the core inner membrane ABC transporter YrbE1A/B-MceG, associated with heterohexamers composed of Mce1A–F or MSMEG_6540/ Mce1B–F. We have demonstrated that complexes containing MSMEG_6540, instead of Mce1A, are, in fact, more important in the newly characterized fatty acid uptake function of Mce1 in *M. smegmatis*. Furthermore, we have discovered a negative regulator of the Mce1 complex, Mce1N, which physically hinders the dominant TMD-NBD interaction (i.e., YrbE1B-MceG) in the ABC transporter. Our work

provides novel insights into the architecture and regulation of mycobacterial Mce complexes.

Based on analogy to other MCE-domain proteins, such as MlaD, PqiB[20] and YebT/LetB[23,24] involved in lipid transport in Gram-negative bacteria (Supplementary Fig. 7a), Mce1A–F and MSMEG_6540/ Mce1B–F likely form hexamers. Each of these Mce proteins contains a single MCE domain (like MlaD) and an extended helical domain (like PqiB), followed by the presence of an additional C-terminal domain of varied sizes. According to AlphaFold2 prediction[26,27,30], Mce1A or MSMEG_6540 can indeed form a heterohexamer with Mce1B–F (Supplementary Fig. 7b). In contrast, a homohexameric model of Mce1A alone could not be generated, agreeing with an earlier observation that the MCE domain of *M. tuberculosis* Mce1A predominantly forms monomers in solution[31]. In the AlphaFold2 model, the heterohexamer adopts an architecture that comprises a hexameric MCE ring connected to a tube formed by a bundle of helical domains; the C-terminal domains of the six Mce proteins organize to form a possible substrate entry point at the very tip (Supplementary Fig. 7b). A continuous hydrophobic tunnel runs through the entire assembly that likely extends across the periplasm to contact the mycobacterial outer membrane (Fig. 6, Supplementary Fig. 7b, c), consistent with its function in lipid uptake. This predicted model resembles the recently solved cryo-electron microscopy (EM) structure of the Mce1 complex remarkably well[32]. Interestingly, *M. smegmatis* possesses MSMEG_6540 as a functional homolog of Mce1A, highlighting the presence of two variants of Mce1 complexes[6], containing Mce1B–F and either Mce1A or MSMEG_6540 (Fig. 6). In support of this idea, removing any other MCE-

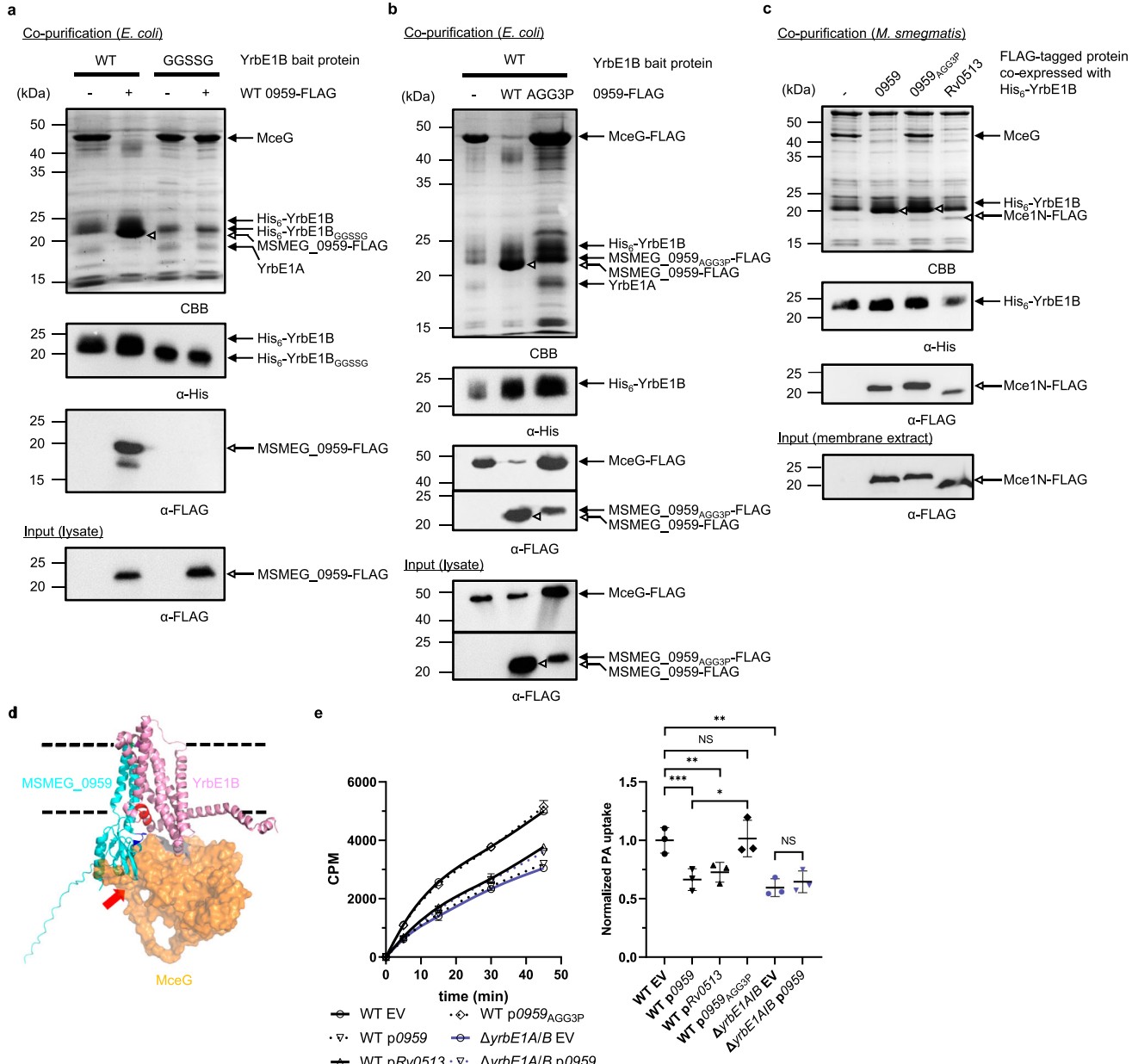

**Fig. 4 | MSMEG_0959 negatively regulates the Mce1 transporter. a** SDS-PAGE and α-His/FLAG immunoblot analyses of proteins affinity-purified from *E. coli* cells over-expressing YrbE1A, MceG, and either wild-type His₆-YrbE1B (WT) or the helix-to-loop mutant His₆-YrbE1B (GGSSG), in the presence (+) or absence (−) of MSMEG_0959-FLAG. **b** SDS-PAGE and α-His/FLAG immunoblot analyses of proteins affinity-purified from *E. coli* cells over-expressing YrbE1A, FLAG-tagged MceG, His-tagged YrbE1B together with FLAG-tagged wild-type MSMEG_0959 (WT) or the loop-rigidified mutant (AGG3P). **c** SDS-PAGE and α-His/FLAG immunoblot analyses of proteins affinity-purified from *M. smegmatis* cells over-expressing His₆-YrbE1B as the bait protein together with MSMEG_0959, the loop-rigidified mutant (0959ₐGG3P) or Rv0513 (Mce1N from *M. tuberculosis*). For (**a**–**c**), the experiments were repeated three times independently with similar results. **d** An overlay of AlphaFold2 models of YrbE1B (pink)·MceG (orange surface) and YrbE1B·MSMEG_0959 (cyan). The helix-to-loop mutation of YrbE1B located on the putative short helix at the YrbE1B-MSMEG_0959 interface is highlighted in red. The rigidified loop on the MSMEG_0959ₐGG3P mutant is highlighted in blue (with an arrowhead). The steric clash between MceG and MSMEG_0959 is indicated by a red arrow. Dashed lines represent arbitrary membrane boundaries based on the hydrophobicity of the transmembrane regions. **e** [¹⁴C]-palmitic acid uptake profiles and rates of indicated *M. smegmatis* strains over-expressing different Mce1Ns from a plasmid grown in the same condition as in (**c**). The uptake profile (left) shows accumulated radioactivity counts in cells over time and is representative of at least three independent experiments. Each data point (mean ± standard deviation) represents results from three technical replicates. CPM count per minute. Uptake rates (right) are quantified based on [¹⁴C]-palmitic acid levels after 30 min incubation. The uptake of individual strains is normalized to that of WT cells harboring the empty vector (WT EV). Mean ± standard deviation of three biological replicates is shown for each group. EV, empty vector (pMV306hsp); *0959*, *MSMEG_0959*. One-way repeated measures ANOVA: NS, not significant; * $p < 0.05$; ** $p < 0.01$, *** $p < 0.0001$. WT EV vs WT p*0959* ($p = 0.0008$), WT EV vs WT p*Rv0513* ($p = 0.0063$), WT EV vs WT p*0959*ₐGG3P ($p = 0.80$), WT p*0959* vs WT p*0959*ₐGG3P ($p = 0.025$), WT EV vs Δ*yrbE1A/B* EV ($p = 0.0068$), Δ*yrbE1A/B* EV vs Δ*yrbE1A/B* p*0959* ($p = 0.11$). The uncropped gels, blots, and raw data underlying **e** are provided in Source Data.

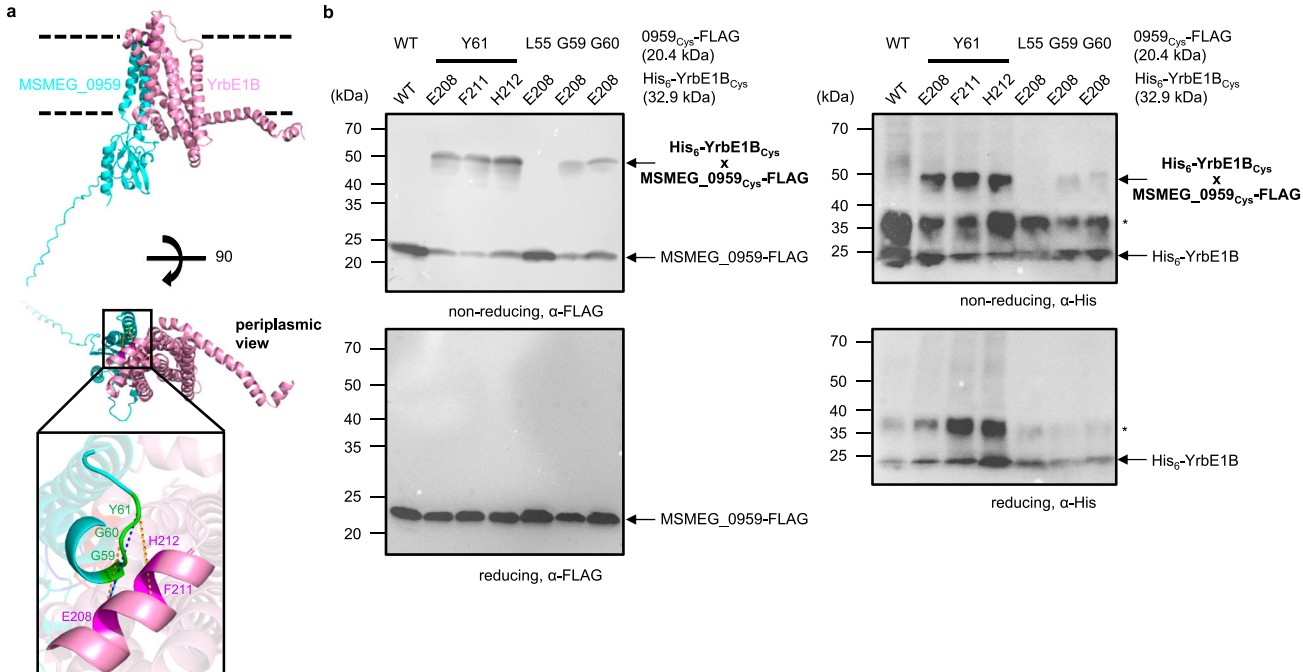

**Fig. 5 | Disulfide crosslinking validates the predicted YrbE1B-MSMEG_0959 model. a** A zoomed-in view of contact sites between YrbE1B and MSMEG_0959 revealed by disulfide crosslinks and mapped onto the AlphaFold2 model. Positions on YrbE1B and MSMEG_0959 that allowed disulfide formation between the two proteins (see (**b**)) are highlighted in magenta and green, respectively. **b** α-FLAG (left) and α-His (right) immunoblot analyses showing crosslinking adducts formed, if any, between MSMEG_0959-FLAG and His$_6$-YrbE1B with cysteines incorporated at

indicated sites when the two proteins were expressed in *E. coli*. Wild-type His$_6$-YrbE1B and MSMEG_0959-FLAG (Lane 1) serve as negative control. The samples were subjected to non-reducing or reducing SDS-PAGE before immunoblotting. A band putatively assigned as His$_6$-YrbE1B dimers, likely due to the hydrophobic nature of the protein, is denoted with an asterisk. This experiment was repeated at least three times independently with similar results. The uncropped blots are provided in Source Data.

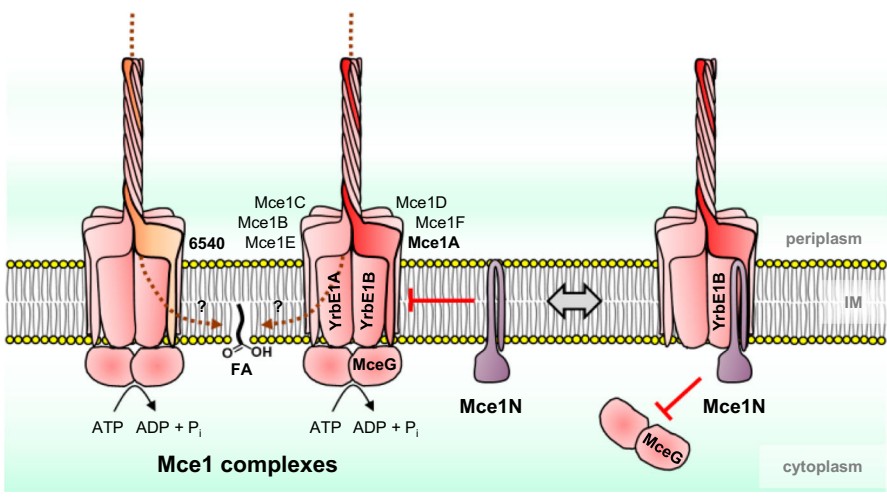

**Fig. 6 | Proposed model of Mce1 complexes and their regulation by Mce1N.** Mce1 complexes are ABC transporters for fatty acids, with YrbE1A/B as TMDs, MceG as NBDs, and heterohexameric MCE-domain proteins as SBPs. There are two variants of Mce1 complexes in *M. smegmatis*, each containing MSMEG_6540 or Mce1A

in addition to Mce1B−F. The relative order of protomers within heterohexameric SBP complexes is indicated based on the AlphaFold2 models (Supplementary Fig. 7b). Mce1N is a negative regulator of the Mce1 system; it specifically binds to YrbE1B and prevents its interaction with MceG.

domain protein in Mce1, but not Mce1A or MSMEG_6540 alone, completely abolishes function in fatty acid uptake−MSMEG_6540 is in fact more important for this process (Fig. 2). Differences between the primary sequences of Mce1A and MSMEG_6540 lie mainly in the extreme C-terminal domain, with differing residues primarily mapped to the substrate entry point in the AlphaFold2 models (Supplementary Fig. 7d). We propose that Mce1 complexes containing either Mce1A or MSMEG_6540 may have distinct substrate specificities, perhaps advantageous for an environmental microorganism like *M. smegmatis*.

Despite the fact that there are known ABC transporters with heterodimeric TMDs, it has not been appreciated that such asymmetry can give rise to major differences in affinity between each TMD and NBD pair−in the case of the Mce1 transporter, the NBD dimer may be predominantly recruited by YrbE1B (Fig. 3c). This might be applicable to other ABC transporters with heterodimeric TMDs since TMD-NBD interfaces of different areas were also observed within some of them (Supplementary Table 2). That TMD-NBD association in the Mce1 transporter relies on the YrbE1B-MceG interaction offers a compelling

reason why the negative regulator Mce1N also selectively binds YrbE1B (Fig. 1c, Supplementary Fig. 6a, b). Based on our modeling and biochemical data (Fig. 4a, d), the cytoplasmic domain of Mce1N likely interacts with YrbE1B extensively at a short helix between the latter's N-terminal amphipathic interface helix and TM1. This short helix is nominally a part of TM1 but is presented at an angle due to a kink introduced by a flexible loop within TM1. Remarkably, this structural feature may be exclusively conserved across YrbEBs, but not YrbEAs (Supplementary Fig. 8). This raises an interesting possibility that beyond the most conserved Mce1 complex, Mce1N may also regulate additional Mce systems by binding to the corresponding YrbEB, perhaps providing a way to redistribute MceG among all Mce systems.

The discovery of a negative regulator that competes with the NBD for binding to the TMDs may be unique to Mce complexes, as known regulatory mechanisms for ABC transporters typically involve modulation of ATPase activity instead of removing the NBD from the complex[19,33–37]. How and when the Mce1 complex is modulated would directly depend on the levels of Mce1N (Fig. 6), which is highly conserved across the *Mycobacterium* genus (present in both fast- and slow-growing, pathogenic and non-pathogenic species). We note that *mce1N* may be in an operon together with four upstream essential genes involved in porphyrin biosynthesis (*hemACDB*). Porphyrin biosynthesis has been reported to increase, suggesting possible upregulation of Mce1N (MSMEG_0959) when *M. smegmatis* cells enter dormancy[38]. In this situation, negative regulation by Mce1N might be a strategy that cells use to inhibit some Mce complexes to conserve ATP. Alternatively, Mce1N levels may correlate with the need for lipid uptake. Outside the host, where fatty acid or related substrates may be scarce, it might be preferable for mycobacterial cells to keep the Mce1 system in a routinely OFF state and avoid futile ATP hydrolysis. During infection, however, cells may require a fully active Mce1 complex to utilize host fatty acids. We do not yet know of any environment-dependent regulation of Mce1N. However, it is noteworthy that in *M. leprae*, which possesses Mce1 as the only Mce system[6], Mce1N (ML2418) lacks the C-terminal cytoplasmic domain (Supplementary Fig. 5b) and is likely non-functional. Combined with the fact that *mce1R*, the gene encoding the transcriptional repressor of the *mce1* operon (Supplementary Fig. 1), is absent in *M. leprae*, the pathogen may have lost its ability to downregulate the Mce1 system, which can facilitate constitutive uptake of fatty acids and related substrates during its obligate parasitic life cycle.

We have uncovered a new regulatory mechanism for Mce1 (Fig. 6) and potentially other Mce systems, which may have implications for how mycobacterial cells coordinate lipid uptake processes. Despite the presence of a transcriptional regulator for some *mce* operons (Supplementary Fig. 1), all Mce transporters apparently share the same ATPase MceG[22]. This indicates that mycobacterial cells have the ability to modulate the levels of each complex yet adjust their activities simultaneously. In the same vein, we speculate that Mce1N could also be a shared negative regulator to counter MceG binding/activation and primarily functions to deactivate all Mce complexes simultaneously. Whether Mce1N has preferences for specific Mce systems remains to be examined. Nevertheless, it likely contributes significantly to the overall regulation toward maintaining a balanced metabolic flux of different lipid substrates required for mycobacterial survival[11,39], which may involve other yet-to-be-identified uptake transporters beyond Mce systems[40].

## Methods
### Plasmid construction
Plasmids used in this study were constructed by traditional ligation or Gibson assembly, with mutations introduced by site-directed mutagenesis if applicable. Primers used for constructing the plasmids are provided in Supplementary Data 2. Backbone plasmids were digested with relevant restriction enzymes (New England Biolabs). The desired DNA fragments to be inserted were amplified by PCR. For traditional ligation, the insert was digested with the same restriction enzymes as the backbone and ligated with the backbone using T4 DNA ligase (New England Biolabs, M0202). For Gibson assembly, desired DNA fragments were inserted into linearized backbones using ClonExpress Ultra One Step Cloning Kit (Vazyme, C115). For site-directed mutagenesis, mutations were introduced to the plasmid using PCR with the original plasmid as the template and primers containing expected changes.

### Construction of *M. smegmatis* knockout strains
*M. smegmatis* strains used in this study are listed in Supplementary Table 3. The knockout strains were constructed by two-step homologous recombination, as reported previously[41]. Briefly, a suicide plasmid pYUB854 with no replication origin and a *hygR* cassette was used as the backbone for the insertion of two flanking regions of the target gene side by side, and a *lacZ-sacB* cassette for blue-white screening and negative selection on sucrose. The plasmid was electroporated at 2250 V using Eporator (Eppendorf) into *M. smegmatis* mc²155 electrocompetent cells, which were subsequently plated on LB plates containing 50 µg/mL hygromycin (Merck, 400051) and 20 µg/mL 5-bromo-4-chloro-3-indolyl-β-D-galactopyranoside (X-gal; 1st BASE, BIO-1020). The plate was incubated at 37 °C for 5–7 days to allow the growth of the first cross-over cells (those forming blue colonies on the plate). Cells harboring the first cross-over at the targeted site were grown in Middlebrook 7H9 broth (Becton Dickinson, 271310) containing 0.5% glycerol and 0.05% tyloxapol (Merck, T0307) and spread onto LB plates containing 20 µg/mL X-gal and 5% sucrose. The plate was again incubated at 37 °C for 4 days to allow the growth of second crossovers (those forming whitish-yellow colonies on the plate). Correct gene deletion mutants were screened by PCR and Sanger sequencing (Bio Basic Asia Pacific) among the second crossovers.

### Pull-down assay in *M. smegmatis*
Plasmids used in this study for protein expression in *M. smegmatis* are listed in Supplementary Table 4. pJEB402 or pMV306hsp vectors carrying genes of interest were transformed into *M. smegmatis* cells by electroporation. The cells were grown in 750 mL tryptic soy broth (TSB; Merck, 1.05459) supplemented with 0.05% Tween 80 (Merck, P8074) to stationary phase ($OD_{600} ~ 5$) and harvested by centrifugation at $4800 \times g$ for 10 min. Harvested cells were resuspended in cold 20 mL TBS (Tris-buffered saline, 20 mM Tris-HCl pH 8.0, 150 mM NaCl) containing 100 µg/mL lysozyme (Merck, L6876), 100 µM phenylmethylsulfonyl fluoride, and 50 µg/mL DNase I, and lysed by two passages through a French Press (GlenMills) at 18,000 psi. The resulting suspension was centrifuged at $4800 \times g$ for 4 min to remove unbroken cells. The supernatant containing the cell lysate was subjected to centrifugation at $95,000 \times g$ for 45 min. The pelleted membrane was resuspended in 5 mL extraction buffer (TBS pH 8.0 containing 5 mM $MgCl_2$, 10% glycerol, 5 mM imidazole, and 1% n-dodecyl-β-D-maltoside (DDM; Anatrace, D310)) and extracted on ice with gentle shaking for 2 h. The suspension was subjected to a second round of centrifugation at $89,000 \times g$ for 45 min. The supernatant was loaded onto 100 µL TALON metal affinity resin (Takara Bio, 635504) pre-equilibrated with TBS pH 8.0 containing 10 mM imidazole. The mixture was allowed to drain by gravity, and the filtrate was loaded back into the column and drained again. The resin was washed with 500 µL washing buffer (20 mM Tris-HCl pH 8.0, 300 mM NaCl, 5 mM $MgCl_2$, 10% glycerol, 8 mM imidazole, and 0.05% DDM) ten times and subsequently eluted with 100 µL elution buffer (TBS pH 8.0 containing 5 mM $MgCl_2$, 10% glycerol, 200 mM imidazole, and 0.05% DDM) for four times. The eluate was concentrated using an Amicon Ultra 10 kDa centrifugal filter (Merck, UFC501096) and kept for further SDS-PAGE analysis. Protein gel sections and bands to be identified were sent to

Taplin Biological Mass Spectrometry Facility, Harvard Medical School (Boston, MA) for capillary LC-MS/MS analysis.

## Protein purification from *E. coli*

Plasmids used in this study for protein expression in *E. coli* are listed in Supplementary Table 5. *E. coli* BL21(λDE3) cells were used as the host strain. Cells were grown to $OD_{600} = 0.6$ in 1.5 L LB broth before 1 mM IPTG was added to induce expression of the proteins for 2 h. The cells were harvested by centrifugation at $4800 \times g$ for 10 min. Harvested cells were resuspended in cold 20 mL TBS (Tris-buffered saline, 20 mM Tris-HCl pH 8.0, 150 mM NaCl) containing 100 µg/mL lysozyme, 100 µM phenylmethylsulfonyl fluoride, and 50 µg/mL DNase I, and lysed by two passages through a French Press at 18,000 psi. The resulting suspension was centrifuged at $4800 \times g$ for 4 min to remove unbroken cells. The supernatant containing the cell lysate was subjected to centrifugation at $95,000 \times g$ for 45 min. The pelleted membrane was resuspended in the 5 mL extraction buffer (TBS pH 8.0 containing 5 mM $MgCl_2$, 10% glycerol, 5 mM imidazole, and 1% DDM) and extracted on ice with gentle shaking for 2 h. The suspension was subjected to a second round of centrifugation at $89,000 \times g$ for 45 min. The supernatant was loaded onto TALON metal affinity resin (Takara Bio) pre-equilibrated with TBS pH 8.0 containing 10 mM imidazole. The mixture was allowed to drain by gravity, and the filtrate was loaded back into the column and drained again. The resin was washed 10 times with the washing buffer (20 mM Tris-HCl pH 8.0, 300 mM NaCl, 5 mM $MgCl_2$, 10% glycerol, 8 mM imidazole, and 0.05% DDM) and subsequently eluted with the elution buffer (TBS pH 8.0 containing 5 mM $MgCl_2$, 10% glycerol, 200 mM imidazole, and 0.05% DDM). The eluate was concentrated using an Amicon Ultra 10 kDa centrifugal filter (MilliporeSigma) and subjected to SDS-PAGE analysis. YrbE1A/B-MceG and YrbE1A/B-MceG$_{K43A}$ were further purified and analyzed by SEC (AKTA, GE Healthcare) at 4 °C on a pre-packed Superdex Increase 200 10/300 GL column (Cytiva). TBS pH 8.0 containing 5 mM $MgCl_2$, 10% glycerol, and 0.05% DDM was used as the eluent. Fractions corresponding to the peaks of our interest were concentrated and subjected to SDS-PAGE analysis.

## SDS-PAGE and immunoblotting

Protein samples were mixed with Laemmli buffer and analyzed by Tris-glycine SDS-PAGE. The gel was visualized by InstantBlue Coomassie Blue Stain (Abcam, ab119211) or subjected to immunoblotting. For the latter, the proteins on the gel were transferred to a polyvinylidene fluoride membrane using a Transblot SD semi-dry transfer system (BioRad) at 25 V for 30 min. The membrane was blocked in 1x casein blocking buffer for at least 1 h. For the detection of His-tagged proteins, a conjugate of mouse monoclonal penta-His antibody and horseradish peroxidase (Qiagen, 34460, 1:5000 dilution) was used. For the detection of FLAG-tagged proteins, a conjugate of mouse monoclonal FLAG-antibody and horseradish peroxidase (Merck, A8592, 1:5000 dilution) was used. To visualize the blots, Luminata Forte Western HRP Substrate (Merck, WBLUF0100) was applied to the membrane, and G:Box chemi XX6 (Synoptics) equipped with GeneSys 1.8 was used to capture the chemiluminescence image.

## Disulfide bond analysis

His$_6$-YrbE1B and MSMEG_0959-FLAG were over-expressed in *E. coli* BL21(λDE3) cells harboring pET28b*his$_6$-yrbE1B* and pCDFduet*MSMEG_0959-FLAG*, with site-specific cysteine substitutions. The cells were grown to $OD_{600} = 0.6$ in LB broth before 1 mM IPTG was added to induce expression of the proteins for 2 h. The cells were harvested by centrifugation at $4000 \times g$ for 3 min. The pellets were resuspended in 2x Laemmeli buffer with (for reducing conditions) or without (for non-reducing conditions) 10% β-mercaptoethanol (Merck, 63689). The samples were centrifuged at $21,000 \times g$ for 10 min before being subjected to SDS-PAGE and immunoblot analyses.

## Palmitic acid uptake assay

The palmitic acid uptake assay was adapted from the one used in ref. 11 *M. smegmatis* cells were grown in the Middlebrook 7H9-glycerol-tyloxapol medium until the stationary phase and subcultured into Sauton's medium with palmitate as the carbon source or TSB for cells over-expressing Mce1Ns. Stationary-phase cells ($OD_{600}$ - 0.4 for cells grown in Sauton and ~5 for those in TSB) were pelleted by centrifugation at $3000 \times g$ for 4 min and resuspended in 1x phosphate-buffered saline (PBS) supplemented with 0.05% Triton X-100 (Merck, T9284) and 0.1% fatty acid-free bovine serum albumin (BSA; Merck, A6003) yielding a final $OD_{600}$ of 0.8. [$^{14}$C]-palmitic acid (Perkin Elmer, NEC075H) was added to a final activity concentration of 0.2 µCi/mL. The cells were incubated at 37 °C with 200 rpm shaking. Cells were harvested at different time points (5, 15, 30, 45 min) by centrifugation at $5000 \times g$ for 1 min from 300 µL suspension and washed once with ice-cold PBS containing Triton X-100 and BSA. The cells were finally resuspended in the same buffer and aliquoted into three plastic scintillation vials for counting using a MicroBeta2 microplate counter equipped with MicroBeta WIW 6.0 (Perkin Elmer). Statistical analyses to compare different samples were done using one-way repeated measures ANOVA with Geisser-Greenhouse correction followed by post hoc Fisher's LSD test in GraphPad Prism 9.5.

## Enzyme-coupled ATPase assay

ATP hydrolytic activity was measured using an NADH-coupled assay[42] adapted for plate readers[43], as previously described[19]. Each 50 µL reaction is made up of 20 mM Tris-HCl pH 8.0, 150 mM NaCl, 5 mM $MgCl_2$, 10% glycerol, 0.05% DDM, 200 mM NADH (Merck, 10128023001), 20 U/mL lactic dehydrogenase (Merck, L1254), 100 U/mL pyruvate kinase (Merck, P9136), 0.5 mM phosphoenolpyruvate (Merck, P7127) and various ATP (Merck, A7699) concentrations. Protein complexes were added to a final concentration of 0.1 µM. Fluorescence emission at 340 nm from the reaction mixture was monitored at 37 °C with 15-s intervals using a Biotek Synergy H1 microplate reader equipped with Biotek Gen5 software (Agilent). A linear fit was performed for the measured fluorescence within 10 min reaction time to obtain the rate of decrease in NADH fluorescence, which was converted to ATP hydrolysis rates using a standard curve obtained with known ADP concentrations. The assay for each sample was performed in technical triplicates, and the data were fit to the built-in Hill equation in OriginPro 2018b.

## Bioinformatic and structural analyses

Topological prediction for MSMEG_0959 was done using the CCTOP server (http://cctop.ttk.hu/). Sequence alignment of proteins was done using pairwise (EMBOSS needle) or multiple sequence alignment (Clustal Omega) tools provided by EMBL-EBI (https://www.ebi.ac.uk/services). Structural prediction for YrbE1A/B-MceG, YrbEA/Bs-MSMEG_0959 was done using ColabFold[28] on the Colaboratory server (Google). Prediction for Mce hexamers was done using the AlphaFold multimer tool on the COSMIC[2] platform[26,30]. For both predictions (Supplementary Data 1), no template information was used, and Amber was used for relaxing the predicted models. Other settings were kept as default. Visualization of structures and overlay of structural models were performed using PyMOL 2.3.0. Interface analysis was performed using ChimeraX 1.4.

## Reporting summary

Further information on research design is available in the Nature Portfolio Reporting Summary linked to this article.

# Data availability

The data that support this study are available from the corresponding author upon request. The source data underlying Figs. 1c, 2, 3a, b, 4e, and Supplementary Figs. 2 and 3 are provided as a Source Data file.

Structural models of protein complexes predicted using AlphaFold2 are provided in Supplementary Data 1. Structural models of individual proteins are available from the AlphaFold Protein Structure Database, https://alphafold.ebi.ac.uk/. Protein structure data used in the course of this work were obtained from Protein Data Bank (PDB) 7CH6; 5UW2; 5UVN; 6MHZ; 7ARM; 7CAG; 2R6G; 4TQU. Source data are provided with this paper.

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

## Acknowledgements
We thank Jia Fu Erh and Wei Shen Ho (National University of Singapore) for constructing plasmids and early affinity purification attempts. We are grateful to Beijing Genomics Institute (BGI) Research for providing DNA synthesis and assembly support. We thank Ross Tomaino (Taplin MS facility, Harvard Medical School) for help with protein identification. This work was supported by Singapore Ministry of Education Academic Research Fund Tier 2 grants MOE2019-T2-1-128 and MOE000116 (S.-S.C.).

## Author contributions
Y.C. and S.-S.C. conceptualized the study, designed the experiments, analyzed the data, and wrote the manuscript. Y.C. performed the experimental studies. Y.W. performed the disulfide crosslinking experiment. S.-S.C. supervised the study.

## Competing interests
The authors declare no competing interests.
