## [Peer Review File · Nature Communications]

A conserved membrane protein negatively regulates Mce1 complexes in mycobacteriaReviewers' Comments:

Reviewer #1:

Remarks to the Author:

This manuscript describes a comprehensive study of Mce1 transporter architecture in *M. smegmatis*. The authors demonstrated for the first time the existence of an assembly mechanism of the Mce1 transporter that is regulated by a protein encoded outside the *mce1* operon and by the availability of the MceG protein. They also demonstrate the existence of a protein, also encoded outside the operon, that would fulfill the same function as Mce1A, among other novel findings. The originality of the results and the quality of the experiments support the publication of this manuscript in *Nature Communication*. However, this reviewer does detect some concerns that should be addressed by the authors.

Lines 102-104. If Mce1A and Msmeg_6540 were functionally redundant, *mce1A* should complement the $\Delta mce1A\Delta 6540$ strain. However, no complementation is observed in the $\Delta mce1A\Delta 6540$ *pmce1A* strain (Figure 2C).

If the introduction of hygromycin cassette in replace to *yrbE1A/B* caused a polar effect in downstream genes of *mce1* operon, as it was suggested by the authors in the legend of Extended Data Figure 2, why the $\Delta mce1A\Delta 6540$ strain could be successfully complemented with the reintroduction of Msmeg_6540 gene (figure 2C)? Also, how do the authors explain that the mutation in *yrbE1A/B* caused a polar effect in the downstream genes that results in a dramatic decrease in the incorporation of palmitic acid but this decrease in palmitic acid uptake was not observed when *mce1A* gene (which is located downstream of *yrbE1A/B*) was replaced by the hygromycin cassette? (Figure 2A). Would these results indicate that there are internal promoters in the *mce1* operon that bypass the transcriptional termination caused by the insertion of the hygromycin cassette?

Figure 2: Biological replicates should be indicated (in addition to technical replicates that are already indicated)

Extended data figure 6: this figure required better explanation. Do the bands of aprox 50kDa in non-reducing condition gel consist on the Msmeg_0959 plus YrbE1B? in this case it should be indicated.

These two articles should be including in the discussion section

1. IUCrJ, 8 (2021), pp. 757-774, 10.1107/S2052252521006199

Structural insights into the substrate-binding proteins Mce1A and Mce4A from *Mycobacterium tuberculosis*

2. doi: 10.1111/mmi.13303. Epub 2016 Feb 5.

An orphaned Mce-associated membrane protein of *Mycobacterium tuberculosis* is a virulence factor that stabilizes Mce transporters

Reviewer #2:

Remarks to the Author:

In this study Chen et al. sought to explore the functions of proteins associated with the Mce1 transport system in mycobacteria. The multiple Mce systems in mycobacteria are homologous to the Mla phospholipid transport pathway in Gram-negative bacteria, and thus far are also associated with the import of lipidic molecules, but are not well understood. Through an array of genetic and biochemical approaches, Chen et al. identified and characterized new proteins outside the Mce1 locus that are required for Mce1-dependent palmitate uptake in *M. smegmatis* and thereby provide novel insights into Mce1 mechanism and regulation. Specifically, they show that MSMEG_6540 is partially redundant with the putative solute-binding protein Mce1A and that MSMEG_0959 (renamed Mce1N) acts as a

negative regulator of Mce1 function, likely by binding preferentially one of the transmembrane domains (YrbE1B) and thus disrupting assembly of the required nucleotide binding domains (MceG). These are noteworthy results, not only because so little is known about Mce systems, but also because the model Chen et al. assemble from their extensive biochemical characterization is novel and will enable more detailed, hypothesis-driven investigations of all Mce systems. The extensive data from well-designed experiments support most of the claims; the more significant comments concern the carbon source-specificity of the growth phenotype and the claim that the multiple substrate-binding proteins Mce1B-F form heterohexamers. Other minor comments concern clarity of presentation, especially for the non-expert.

General question: There appears to be palmitate uptake at low level even in the “negative control” strains lacking yrbE1 and mce1F (Figure 2). Could the authors please comment on what this activity represents? Could there be another uptake system? Or is this non-facilitated uptake? Or absorption?

The authors conclude based on data in Figure 2B that Mce1 is specific for palmitate, ostensibly by showing a growth phenotype on palmitate vs. glycerol as a sole carbon source. First, the experiments were performed in different media (Sauton/palmitate vs. 7H9/glycerol). The phenotype on Sauton/palmitate is weak (differences in OD600 of ~0.1) and growth is more than an order of magnitude better supported on 7H9/glycerol, but similar patterns among the strains may be apparent (i.e., OD600 differences of <1; the data on glycerol are difficult to discern--even when zoomed on a screen-- and we recommend showing both graphs in full size). From the methods, it is not clear what the exact composition of the 7H9-based medium is, and thus if glycerol is indeed a sole carbon source under these conditions. However, this point re: carbon source specificity is not central to this study or necessary to support later conclusions. We therefore suggest that the authors either perform additional experiments under consistent conditions to support this point rigorously or remove it from the study.

Re: methods and 7H9 medium more generally, please be clear if you use 7H9 to refer solely to the Middlebrook 7H9 base. Not all studies do this (some use “7H9” as shorthand for 7H9 with various additives such as albumin-dextrose-catalase).

Also, general comment for all of Fig. 2 and for Fig. 4F to please provide analysis of significance with appropriate statistical tests.

The authors conclude that Mce1B-F form heterohexamers. The basis for this appears to be the equivalent loss of palmitate uptake in the Δ mce1F/mam1 and Δ yrbE1 mutants (Figure 2D). However, the authors do not test how the loss of mce1B-E affects uptake. This would seem to leave open the possibility that while Mce1B-F all interact with YrbE1 (Figure 1), this represents interactions with homohexamers of each and while the Mce1F homohexamer is important for function, those formed by Mce1B-E are relatively unimportant, at least under the conditions tested. Further, there appears to be precedent only for homohexamers, based on the examples of PqiB and YebT/LetB provided by the authors (p. 3, lines 55-56). We suggest that the authors address this possibility.

Methods/reproducibility: While some primers are provided, no cloning methods are described. Please provide at least brief and sufficient description about cloning strategies as relevant to the constructs (e.g., all by traditional digestion/ligation?). In Table S2 please also list suicide plasmids for the knockout strains. Also please provide mutagenic primers in Table S3.

More minor comments regarding accuracy and clarity of presentation:

General comment: To the best of our knowledge, the authors are the first to shown that palmitate uptake depends on Mce1 in *M. smegmatis* (vs. previous studies in *Mtb*), showing conservation of function. The authors are encouraged to add this point early on.

Also, authors provide a degree of inference/speculation about Mce1N, but less so for MSMEG_6540. Are there other "orphan" Mce SBP homologues like MSMEG_6540 (i.e., not associated with an mce operon) in Msm? Other mycobacteria? A minor suggestion to add relevant discussion.

Figure 4E shows only the model; we suggest that it is important to show here the data that supports it. Suggest moving Extended Data Fig. 6 into the main figure (and then possibly making this a separate figure).

Inevitably, this is a lot of alphabet soup with so many proteins. We suggest putting a model of the core Mce1 proteins (similar to the middle complex in Fig. 5) into Fig. 1 (per comments above regarding homo vs. heterohexamers, perhaps pointing to the SBP hexamer and simply listing Mce1A-F instead of assigning them to particular monomers) and labeling the NBD, TBD, and SBP components as groups. This would provide a very welcome front-end reference as the reader navigates the use of individual protein names and NBD/TBD/SBP terminology used throughout.

Lines 15, 36. Suggest rewording to account for the fact that Mce systems characterized *so far* appear to be involved in uptake of lipidic substrates (fatty acid, cholesterol). Encourage being clear that there is a lot of room for clarification (i.e., whether these are the only substrates of Mce 1, 3; also what Mce2, 4 do).

Also, in line 36, the term "virulence factors" may not be the most accurate term to designate this group given that the following sentence emphasizes its ubiquity across the genus (including non-pathogenic species). Suggest rewording.

Line 30: Latent TB is a clinical state, not an infectious agent, so suggest that "one quarter... *have* latent TB" rather than "are carriers of".

Lines 74-75. What do the authors mean by "this was done in a strain that presumably up-regulated the mce1 operon due to the lack of Mce1R"? Is this done in a special background strain? Or does the parent strain of *M. smegmatis* used naturally lack Mce1R? Or does the Msm genome lack Mce1R? Please clarify.

Lines 112-114. What was or was not possible is not clear as written. What was not successful—overexpression (was the operon toxic)? Or purification of the complex? Or both? Please amend.

Nomenclature. We believe the standard capitalization for Msm genes/proteins is MSMEG_#### rather than Msmeg_####. Please amend.

Responses to reviewer comments (comments in BLACK; responses in BLUE)

Reviewer #1 (Remarks to the Author):

This manuscript describes a comprehensive study of Mce1 transporter architecture in *M. smegmatis*. The authors demonstrated for the first time the existence of an assembly mechanism of the Mce1 transporter that is regulated by a protein encoded outside the *mce1* operon and by the availability of the MceG protein. They also demonstrate the existence of a protein, also encoded outside the operon, that would fulfill the same function as Mce1A, among other novel findings. The originality of the results and the quality of the experiments support the publication of this manuscript in Nature Communication. However, this reviewer does detect some concerns that should be addressed by the authors.

Lines 102-104. If Mce1A and Msmeg_6540 were functionally redundant, *mce1A* should complement the $\Delta mce1A\Delta 6540$ strain. However, no complementation is observed in the $\Delta mce1A\Delta 6540$ *pmce1A* strain (Figure 2C).

We thank the reviewer for pointing this out. We were also initially puzzled by this lack of complementation. After careful re-examination of the initial constructs used to generate $\Delta mce1A$, we noticed that the putative ribosome binding site for *mce1B* (downstream of *mce1A*) was not fully preserved after the deletion of *mce1A*. This might have led to a mild polar effect on the rest of the operon, which cannot be complemented by expressing *mce1A*. To eliminate this possible effect, we have now reconstructed new $\Delta mce1A$ and $\Delta mce1A\Delta 6540$ strains. Interestingly, removing *mce1A* alone causes an increase in PA uptake over WT cells, while doing so in the absence of *MSMEG_6540* reduces PA uptake to the basal level seen in $\Delta yrbE1A/B$ cells. These changes can now be complemented by ectopic expression of *mce1A*. The new data show that Mce1A and *MSMEG_6540* are functionally redundant. We believe that complexes with Mce1A are much less effective in PA uptake than those with *MSMEG_6540*; when *mce1A* is deleted, increased PA uptake is likely contributed by more Mce1 complexes containing *MSMEG_6540*. We have included these new results and modified the text in the revised manuscript.

If the introduction of hygromycin cassette in replace to *yrbE1A/B* caused a polar effect in downstream genes of *mce1* operon, as it was suggested by the authors in the legend of Extended Data Figure 2, why the $\Delta mce1A\Delta 6540$ strain could be successfully complemented with the reintroduction of *Msmeg_6540* gene (figure 2C)? Also, how do the authors explain that the mutation in *yrbE1A/B* caused a polar effect in the downstream genes that results in a dramatic decrease in the incorporation of palmitic acid but this decrease in palmitic acid uptake was not observed when *mce1A* gene (which is located downstream of *yrbE1A/B*) was replaced by the hygromycin cassette? (Figure 2A). Would these results indicate that there are internal promoters in the *mce1* operon that bypass the transcriptional termination caused by the insertion of the hygromycin cassette?

We have generated all our strains using an unmarked deletion strategy, i.e., the chromosomal gene is either removed with no scar or replaced by a *HindIII* restriction site (further detailed in the Methods section). Using this strategy, we have now additionally generated mutant strains lacking any of *mce1A*–

F, to further support our claim that Mce1A–F form a heterohexamer (see Reviewer 2 comments), but also allowing better assessment of polar effects within the *mce1* operon. We found that deleting different *mce* genes resulted in different levels of polar effects. Strains lacking *mce1A*, *mce1C*, or *mce1D* can be fully complemented with ectopic expression of the removed gene alone, indicating no polar effects. For the $\Delta yrbE1A/B$ and $\Delta mce1F$ strains, we have already shown that there was essentially no complementation conferred by expression of *yrbE1A/B* or *mce1F* alone, consistent with strong polar effects. Finally, the $\Delta mce1B$ or $\Delta mce1E$ strains could be partially complemented by expression of *mce1B* or *mce1E* respectively, but fully complemented by *mce1B* or *mce1E* together with all the downstream genes (*mce1C–mam1D* or *mce1F–mam1D*), indicating milder polar effects somehow caused by these deletions. In summary, it is not clear why deleting specific genes in the *mce1* operon gave distinct polar effects, if any. Such effects may depend on local arrangement of sequences that can affect transcription and/or translation.

Figure 2: Biological replicates should be indicated (in addition to technical replicates that are already indicated)

Thanks for the suggestion. We have now obtained results from biological triplicates for the palmitic acid uptake assay and performed statistical analyses. We have included these results in the new Figure 2 and Figure 4e.

Extended data figure 6: this figure required better explanation. Do the bands of approx 50kDa in non-reducing condition gel consist on the Msmeg_0959 plus YrbE1B? in this case it should be indicated.

Yes, the 50 kDa bands correspond to the crosslinked adducts of YrbE1B and MSMEG_0959. We have annotated the protein adduct with bold fonts as “**YrbE1BxMSMEG_0959**”. We have also moved this figure into the main text as Figure 5b.

These two articles should be including in the discussion section

1. IUCrJ, 8 (2021), pp. 757-774, 10.1107/S2052252521006199

Structural insights into the substrate-binding proteins Mce1A and Mce4A from Mycobacterium tuberculosis

2. doi: 10.1111/mmi.13303. Epub 2016 Feb 5.

An orphaned Mce-associated membrane protein of Mycobacterium tuberculosis is a virulence factor that stabilizes Mce transporters

Both references have been added. The first paper is cited in the Discussion section and the second one in the Results section.

Reviewer #2 (Remarks to the Author):

In this study Chen et al. sought to explore the functions of proteins associated with the Mce1 transport system in mycobacteria. The multiple Mce systems in mycobacteria are homologous to the Mla phospholipid transport pathway in Gram-negative bacteria, and thus far are also associated with the import of lipidic molecules, but are not well understood. Through an array of genetic and biochemical approaches, Chen et al. identified and characterized new proteins outside the Mce1 locus that are required for Mce1-dependent palmitate uptake in *M. smegmatis* and thereby provide novel insights into Mce1 mechanism and regulation. Specifically, they show that MSMEG_6540 is partially redundant with the putative solute-binding protein Mce1A and that MSMEG_0959 (renamed Mce1N) acts as a negative regulator of Mce1 function, likely by binding preferentially one of the transmembrane domains (YrbE1B) and thus disrupting assembly of the required nucleotide binding domains (MceG). These are noteworthy results, not only because so little is known about Mce systems, but also because the model Chen et al. assemble from their extensive biochemical characterization is novel and will enable more detailed, hypothesis-driven investigations of all Mce systems. The extensive data from well-designed experiments support most of the claims; the more significant comments concern the carbon source-specificity of the growth phenotype and the claim that the multiple substrate-binding proteins Mce1B-F form heterohexamers. Other minor comments concern clarity of presentation, especially for the non-expert.

General question: There appears to be palmitate uptake at low level even in the “negative control” strains lacking *yrbE1* and *mce1F* (Figure 2). Could the authors please comment on what this activity represents? Could there be another uptake system? Or is this non-facilitated uptake? Or absorption?

The residual palmitate uptake we have observed is consistent with the fact that these strains could still grow using palmitate as the sole carbon source. Similar results have been shown in previous studies done in *M. tuberculosis*. It is not clear how fatty acid uptake is mediated in mycobacteria when Mce1 is absent. However, we note that an alternative transporter has previously been suggested but not studied in detail (PMID 29360453). Non-facilitated uptake is also a possibility.

The authors conclude based on data in Figure 2B that Mce1 is specific for palmitate, ostensibly by showing a growth phenotype on palmitate vs. glycerol as a sole carbon source. First, the experiments were performed in different media (Sauton/palmitate vs. 7H9/glycerol). The phenotype on Sauton/palmitate is weak (differences in OD600 of ~0.1) and growth is more than an order of magnitude better supported on 7H9/glycerol, but similar patterns among the strains may be apparent (i.e., OD600 differences of <1; the data on glycerol are difficult to discern--even when zoomed on a screen-- and we recommend showing both graphs in full size). From the methods, it is not clear what the exact composition of the 7H9-based medium is, and thus if glycerol is indeed a sole carbon source under these conditions. However, this point re: carbon source specificity is not central to this study or necessary to support later conclusions. We therefore suggest that the authors either perform additional experiments under consistent conditions to support this point rigorously or remove it from the study. We agree with your advice and have decided to remove the growth curves from the manuscript.

Re: methods and 7H9 medium more generally, please be clear if you use 7H9 to refer solely to the

Middlebrook 7H9 base. Not all studies do this (some use “7H9” as shorthand for 7H9 with various additives such as albumin-dextrose-catalase).

7H9 was intended to refer to the Middlebrook 7H9 base. We have modified the text to make this clear.

Also, general comment for all of Fig. 2 and for Fig. 4F to please provide analysis of significance with appropriate statistical tests.

We have repeated the experiments with biological triplicates and performed statistical analyses of the results. We have included these in the new figures.

The authors conclude that Mce1B-F form heterohexamers. The basis for this appears to be the equivalent loss of palmitate uptake in the $\Delta mce1F/mam1$ and $\Delta yrbE1$ mutants (Figure 2D). However, the authors do not test how the loss of mce1B-E affects uptake. This would seem to leave open the possibility that while Mce1B-F all interact with YrbE1 (Figure 1), this represents interactions with homohexamers of each and while the Mce1F homohexamer is important for function, those formed by Mce1B-E are relatively unimportant, at least under the conditions tested. Further, there appears to be precedent only for homohexamers, based on the examples of PqiB and YebT/LetB provided by the authors (p. 3, lines 55-56). We suggest that the authors address this possibility.

We thank the reviewer for raising this point. We have now additionally generated and tested mutants lacking any of *mce1B-E*. After polar effects, if any, were alleviated, the mutants all demonstrated basal level palmitic acid uptake equivalent to that of the $\Delta yrbE1A/B$ mutant, consistent with a complete loss of Mce1 function. These data corroborate our conclusion that Mce1A/MSMEG_6540 forms a heterohexamer together with Mce1B-F. We have included these new results and modified the text in the revised manuscript.

Methods/reproducibility: While some primers are provided, no cloning methods are described. Please provide at least brief and sufficient description about cloning strategies as relevant to the constructs (e.g., all by traditional digestion/ligation?). In Table S2 please also list suicide plasmids for the knockout strains. Also please provide mutagenic primers in Table S3.

We have added a section for plasmid construction in Methods. The plasmid list and the primer list have also been updated accordingly.

More minor comments regarding accuracy and clarity of presentation:

General comment: To the best of our knowledge, the authors are the first to shown that palmitate uptake depends on Mce1 in *M. smegmatis* (vs. previous studies in *Mtb*), showing conservation of function. The authors are encouraged to add this point early on.

We thank the reviewer for highlighting this. We have added this point in the final paragraph of Introduction and in the Discussion section.

Also, authors provide a degree of inference/speculation about Mce1N, but less so for MSMEG_6540. Are there other “orphan” Mce SBP homologues like MSMEG_6540 (i.e., not associated with an mce operon) in Msm? Other mycobacteria? A minor suggestion to add relevant discussion.

We had already included discussion suggesting that MSMEG_6540 and Mce1A provide distinct substrate specificities in the Mce1 complexes. While there is a third Mce1A homolog in *M. smegmatis*, namely MSMEG_5818, it was not co-purified with YrbE1B, nor was it highly similar to other Mce proteins. This makes it difficult to infer its function. The other example of a Mce1A homolog is MAP_3289c in *M. paratuberculosis* (PMID 17324287), with 92% sequence identity. The limited differences between the two homologs are however mainly located at the N-termini of the proteins, which are unlikely to affect substrate specificity. There is no other known orphaned Mce SBP homolog in mycobacteria. Given the limited information about these Mce1A homologs, we have decided not to add further speculation.

Figure 4E shows only the model; we suggest that it is important to show here the data that supports it. Suggest moving Extended Data Fig. 6 into the main figure (and then possibly making this a separate figure).

Thanks for the suggestion. We have put the original Extended Data Fig. 6 as Figure 5b, together with the original Figure 4E now as Figure 5a.

Inevitably, this is a lot of alphabet soup with so many proteins. We suggest putting a model of the core Mce1 proteins (similar to the middle complex in Fig. 5) into Fig. 1 (per comments above regarding homo- vs. hetero-hexamers, perhaps pointing to the SBP hexamer and simply listing Mce1A-F instead of assigning them to particular monomers) and labeling the NBD, TBD, and SBP components as groups. This would provide a very welcome front-end reference as the reader navigates the use of individual protein names and NBD/TBD/SBP terminology used throughout.

We thank the reviewer for this advice. We have added such an Mce1 model as new Figure 1b.

Lines 15, 36. Suggest rewording to account for the fact that Mce systems characterized *so far* appear to be involved in uptake of lipidic substrates (fatty acid, cholesterol). Encourage being clear that there is a lot of room for clarification (i.e., whether these are the only substrates of Mce 1, 3; also what Mce2, 4 do).

We agree with this point. We have modified sentences in the abstract and Introduction to accommodate the possibility of unidentified Mce substrates of different nature.

Also, in line 36, the term “virulence factors” may not be the most accurate term to designate this group given that the following sentence emphasizes its ubiquity across the genus (including non-pathogenic species). Suggest rewording.

We have adjusted the order of the paragraph to avoid the potential confusion.

Line 30: Latent TB is a clinical state, not an infectious agent, so suggest that “one quarter... *have* latent TB” rather than “are carriers of”.

Thanks for pointing this out. We have changed the expression.

Lines 74-75. What do the authors mean by “this was done in a strain that presumably up-regulated the *mce1* operon due to the lack of Mce1R”? Is this done in a special background strain? Or does the parent strain of *M. smegmatis* used naturally lack Mce1R? Or does the Msm genome lack Mce1R? Please clarify.

We have revised the sentence to avoid the ambiguity. “This was done in a $\Delta mce1R$ strain, where the *mce1* operon is presumably up-regulated.”

Lines 112-114. What was or was no possible is not clear as written. What was not successful—overexpression (was the operon toxic)? Or purification of the complex? Or both? Please amend.

We failed to purify the whole complex due to low expression levels of MCE-domain proteins in *E. coli*. We have modified the text to clarify this point. “We attempted but failed to over-express the *mce1* operon along with *mceG* in *E. coli*. Expression levels of Mce1A–F were very low, which precluded purification of any stable Mce1 complex.”

Nomenclature. We believe the standard capitalization for Msm genes/proteins is MSMEG_#### rather than Msmeg_####. Please amend.

We thank the reviewer for pointing this out. We have changed all the names accordingly.

Reviewers' Comments:

Reviewer #1:

Remarks to the Author:

The authors have addressed the comments and concerns and have included new experiments that further support the study's conclusions. The manuscript should be accepted for publication.

Minor comments:

1. Please, explain what p1C and p1D mean in figure 2.
2. The band at lane Y61/E208 in figure 5 (right panel) is almost indiscernible.
3. Line 574, include in the text some explanation of the generation of YrbE1B dimers.

Reviewer #2:

Remarks to the Author:

In this revised manuscript, Chen et al. overall satisfactorily addressed comments from reviewers and we appreciate the effort required for the numerous additional strains and experiments. The only remaining concern that requires further attention is the conclusion that Mce1A and MSMEG_6540 are functionally redundant. The additional data provided in Figure 2 underscore that the situation is complicated and that the systems as created (KO and complement) may not be sufficient to fully distinguish the relationship between these two genes. First, *mce1A* KO is now shown to have a reproducible *increase* in PA uptake. These data alone would not support a role for *mce1A* KO in mediating PA uptake and thus Mce1A function is then supported solely by homology and its position within the *mce1* locus. An alternate explanation that is still consistent with the observed phenotype is that the increased uptake results from adaptation to the genetic modification—e.g., upregulation of MSMEG_6540. The authors could consider raising this possibility. In contrast, MSMEG_6540 KO produces a much more marked phenotype with loss of PA uptake. Interpreting the complementation phenotypes is complicated by the fact that the strong constitutive MOP promoter (rather than native promoters) was used for the complements. This could lead to non-physiological expression and corresponding non-physiological complement phenotypes. Indeed, complementing the MSMEG_6540 single KO by overexpressing Mce1A might more clearly illustrate redundant function than the single complements of the double KO shown in Fig. 2F. However, overexpression of Mce1A on the WT background as shown in Figure S3 *decreases* PA uptake, which along with data in Fig. 2A and 2F showing increases in PA uptake in the *mce1A* KO rather supports Mce1A as a negative regulator of uptake. Generally, based on the data as shown, the most strongly supported conclusion (i.e., disregarding statistically significant, but comparatively small changes in uptake) is that Mce1A and MSMEG_6540 have distinct roles, and indeed that for palmitic acid uptake, Mce1A is not required and MSMEG_6540 fulfills the necessary function. While we do not feel that additional experiments are necessary, we strongly recommend additional/modified, nuanced discussion of the unexpected and somewhat contradictory observations as noted above and that the authors should reconsider their main conclusion that Mce1A and MSMEG_6540 are functionally redundant, in favor of more qualified statements on this point throughout the manuscript. (Additional minor comment: Welch's t-test is used throughout, including in instances where multiple comparisons are made and where a method that mitigates risk for Type I error, such as 1-way ANOVA with appropriate correction, is likely called for.)

Responses to reviewer comments (comments in BLACK; responses in BLUE)

Reviewer #1 (Remarks to the Author):

The authors have addressed the comments and concerns and have included new experiments that further support the study's conclusions. The manuscript should be accepted for publication.

Minor comments:

1. Please, explain what p1C and p1D mean in figure 2.

We have added brief explanations in the figure legend.

2. The band at lane Y61/E208 in figure 5 (right panel) is almost indiscernible.

Thanks for pointing this out. We have repeated the experiment and provided new blots for this figure.

3. Line 574, include in the text some explanation of the generation of YrbE1B dimers.

We routinely observed bands consistent with YrbE1B dimers on gels and blots, likely due to the hydrophobic nature of the protein. We have added this information in the figure legend.

Reviewer #2 (Remarks to the Author):

In this revised manuscript, Chen et al. overall satisfactorily addressed comments from reviewers and we appreciate the effort required for the numerous additional strains and experiments. The only remaining concern that requires further attention is the conclusion that Mce1A and MSMEG_6540 are functionally redundant. The additional data provided in Figure 2 underscore that the situation is complicated and that the systems as created (KO and complement) may not be sufficient to fully distinguish the relationship between these two genes. First, *mce1A* KO is now shown to have a reproducible *increase* in PA uptake. These data alone would not support a role for *mce1A* KO in mediating PA uptake and thus Mce1A function is then supported solely by homology and its position within the *mce1* locus. An alternate explanation that is still consistent with the observed phenotype is that the increased uptake results from adaptation to the genetic modification—e.g., upregulation of MSMEG_6540. The authors could consider raising this possibility. In contrast, MSMEG_6540 KO produces a much more marked phenotype with loss of PA uptake. Interpreting the complementation phenotypes is complicated by the fact that the strong constitutive MOP promoter (rather than native promoters) was used for the complements. This could lead to non-physiological expression and corresponding non-physiological complement phenotypes. Indeed, complementing the MSMEG_6540 single KO by overexpressing Mce1A might more clearly illustrate redundant function than the single complements of the double KO shown in Fig. 2F. However, overexpression of Mce1A on the WT background as shown in Figure S3 *decreases* PA uptake, which along with data in Fig. 2A and 2F showing increases in PA uptake in the *mce1A* KO rather supports Mce1A as a negative regulator of uptake. Generally, based on the data as shown, the most strongly supported conclusion (i.e., disregarding statistically significant, but comparatively small changes in uptake) is that Mce1A and MSMEG_6540 have distinct roles, and indeed that for palmitic acid uptake, Mce1A is not required and MSMEG_6540 fulfills the necessary function. While we do not feel that additional experiments are necessary, we strongly recommend additional/modified, nuanced discussion of the unexpected and somewhat contradictory observations as noted above and that the authors should reconsider their main conclusion that Mce1A and MSMEG_6540 are functionally redundant, in favor of more qualified statements on this point throughout the manuscript. (Additional minor comment: Welch's t-test is used throughout, including in instances where multiple comparisons are made and where a method that mitigates risk for Type I error, such as 1-way ANOVA with appropriate correction, is likely called for.)

We overall agree with the reviewer that more explicit and nuanced discussion of the roles of Mce1A and MSMEG_6540 would be useful. Our belief that there is some level of functional redundancy was guided by the observations that (1) loss of MSMEG_6540 alone did not fully abolish PA uptake (in Δ MSMEG_6540, Mce1A was presumably expressed at the native level), and (2) ectopic expression of Mce1A recovered PA uptake of the Δ *mce1A* Δ 6540 strain to the levels observed in Δ MSMEG_6540. However, as the reviewer correctly pointed out, the difference with/without Mce1A is small (though statistically significant), suggesting that Mce1A can contribute to fatty acid uptake but that is not the main role. Mce1A may instead be more important for uptake of other substrates as we have discussed in the text. We have modified the text to emphasize on the idea that Mce1A has a

distinct role.

It is also true that (1) loss of Mce1A alone increases PA uptake and (2) overexpressing Mce1A in WT cells reduces PA uptake. In these contexts where MSMEG_6540 is present, it may appear that Mce1A is a negative regulator. However, since Mce1A and MSMEG_6540 must function as part of the Mce1 complexes and therefore share the common components, a more consistent idea would be that Mce1A and MSMEG_6540 compete to form a full Mce1 complex, and that the levels of Mce1A would negatively correlate with the levels of Mce1 complexes containing MSMEG_6540, impacting according to what we have observed. Furthermore, the fact that other mycobacteria only contain Mce1A (no other homolog), the idea of Mce1A serving as a negative regulator of the Mce1 complex is highly unlikely. We believe and agree with the reviewer that Mce1A has a distinct role, perhaps in uptake of other substrates. We have modified the text to reflect this point.

We have now used one-way repeated measures ANOVA with Geisser-Greenhouse correction followed by post hoc Fisher's LSD test for our statistical analyses.